# The Intrinsic and Instrumental Values of Blockchain to Provide Beef Traceability in Hong Kong, South Korea, and the United States

**DOI:** 10.3390/foods12234209

**Published:** 2023-11-22

**Authors:** Jisung Jo, Jayson L. Lusk

**Affiliations:** 1Logistics and Maritime Industry Research Department, Korea Maritime Institute, 26 Haeyang-ro 301beon-gil, Busan 49111, Republic of Korea; 2Division of Agricultural Science & Natural Resources, Oklahoma State University, 139 Agriculture Hall, Stillwater, OK 74078, USA; jayson.lusk@okstate.edu

**Keywords:** blockchain, intrinsic and instrumental value, beef, traceability, WTP, consumer preferences

## Abstract

Although previous research has identified that consumers are willing to pay for traceability, it remains unknown which types of traceability information might have the highest value, and whether consumers have an intrinsic value for blockchain technology above and beyond the instrumental value of providing traceability. A choice experiment was conducted with over 1500 consumers in Hong Kong, South Korea, and the U.S. In all three countries, consumers were willing to pay premiums for beef with traceability related to all parts of the supply chain, country of origin, and temperature history; however, the preference ordering of beef from different countries varied across Hong Kong, South Korea, and the U.S. The intrinsic value of using blockchain to deliver traceability information differed by country and by attribute, and consumers in the U.S. were most sensitive to the information describing blockchain technology. Even when traceability conveys negative information, such as temperature rising above safe levels for a short period, we find that consumers prefer knowing to not knowing, suggesting uncertainty and ambiguity aversion.

## 1. Introduction

Globally, 600 million people (almost 1 in 10 people in the world) fall ill, and 420,000 die from foodborne diseases, every year (WHO [1]). Foodborne illnesses are caused by consuming food contaminated with bacteria, viruses, parasites, or chemical substances. Contamination can occur at any stage of the food supply chain (WHO [1]). The food supply chain is globally connected, and disruptions in the supply chain have global impacts that extend beyond the country level. As disruptions in the food supply chain and the resulting health issues continue to emerge, interest in food traceability systems has increased. Food traceability allows consumers to access information on every node (farm, manufacturer, transporter, importer, retailers, etc.), as well as cold chain management information (temperature, humidity, vibration, etc.) for each node.

Badia-Melis et al. [2] pointed out the vulnerabilities of the current food traceability systems stemming from the inability to link food chain records, inaccuracy and errors in the records, and delays in obtaining essential data. They argue that technological innovations might be a solution to the current issues and to enhance the efficiency of food traceability systems. Magalhães et al. [3] also estimated a high degree of correlation between the evolution of food traceability technology and food outbreaks in the United States and Germany, indicating that traceability technology is one of the critical factors for reducing the number of food outbreaks.

Blockchain is a relatively new technology that has a potential application in food traceability. Representative characteristics of blockchain technology include disintermediation, transparency, decentralization, and immutability (Bischoff and Seuring [4]). Among them, the immutability of blockchain, which ensures tamper-proof data-sharing, is key to resolving the old problems of food fraud and quality. Blockchain applications in the food industry are rapidly growing. The estimated size of the blockchain market for agriculture and the food supply chain was $133.4 million in 2020 and it is projected to reach $948.6 million in 2025 (Markets and Markets [5]). The compound annual growth rate (CAGR) from 2019 to 2025 is estimated to be as high as 48.1%. There are four different segments of blockchain applications: product traceability, payments and settlements, smart contracts, and governance, risk and compliance management (Market and Market [5]). The market of ‘product traceability, tracking and visibility’ was the largest in 2020, and is projected to be the largest in 2025 as well.

Despite the rapid growth in the applications of blockchain technology to food supply chains, it remains unknown whether, and to what extent, consumers value the technology. Previous research has identified that consumers are willing to pay more for traceability (Dickinson et al. [6]; Dickinson and Bailey [7]; Van et al. [8]; Loureiro and Umberger [9]; Shew et al. [10]; Lin et al. [11]). Some companies have advertised the use of blockchain technology to improve their food supply chains, suggesting a belief that the consumers might care about the means in which traceability is conveyed.

### 1.1. Literature Review on Blockchain Technology Applications and the Impact on the Food Supply Chain

An increasing number of studies have been conducted on blockchain applications in food traceability systems. Feng et al. [12] showed an upward trend in the publications related to the applications of blockchain for food traceability from 2005 to 2019, with a sharp uptick since 2016. Feng et al. [12] suggested the architecture design of a blockchain-based food traceability system, which consists of four layers—the business layer, the IoT traceability layer, the blockchain layer, and the application layer. Iftekhar and Cui [13] mentioned that the COVID19 virus could spread along the import and export food supply chain. They suggested a blockchain-enabled supply chain architecture, consisting of an application layer, a blockchain layer, and a physical layer to ensure food safety and tamper-proof data sharing. Bumblauskas et al. [14] conducted a proof of concept to investigate the implementation of blockchain in the egg supply chain from farms to consumers in the U.S. They showed that blockchain can supplement food traceability in new and existing supply chains, and that blockchain applications are possible without immense effort from stakeholders. Vivaldini [15] also conceptualized blockchain applications in food service distribution, and presented how and where blockchain can be applied by each distributor operation. Behnke and Janssen [16] investigated four different dairy food supply chain processes and suggested the boundary conditions for traceability and blockchain. They emphasized the importance of standardizing the traceability processes and interfaces before utilizing blockchain technology. Cao et al. [17] proposed a blockchian based multisignature approach to digitally transform supply chain governance in multi-food supply chains, particularly in the beef industry. They conducted an exploratory case study, and the real-case implementation illustrates the effectiveness of the blockchain-based multisignature approach in enhancing digital governance within the beef supply chain.

Some attempts have been made to estimate the impact of blockchain applications on the food supply chain in terms of firm performance. Stranieri et al. [18] conducted semi-structured interviews with key managers who have blockchain adoption experience in three supply chains (poultry, lemons, and oranges). They found that blockchain has a positive impact on profit, extrinsic food quality attributes, information accessibility, and availability. Duan et al. [19] utilized a content analysis based on the literature review and proposed the benefits of blockchain adoption in the food supply chain; improving food traceability, information transparency, and recall efficiency. Sheel and Nath [20] conducted an extensive literature review on blockchain and showed that adopting the blockchain technology may improve the adaptability, alignment, and agility of the supply chain. Sander et al. [21] investigated the potential acceptance of blockchain technology as a viable transparency and traceability system for meat. Their study revealed that the blockchain technology has a significant positive impact on consumers’ purchasing decisions. These insights were gleaned from semi-structured interviews conducted with seven retail managers, four government officials, and one blockchain service provider. These studies are qualitative, perhaps because blockchain has only recently been introduced to the food industry.

### 1.2. Literature Review on Consumer Preferences for Blockchain Applications on Beef Traceability

Some researchers have studied consumer perceptions and the demand for blockchain applications. Rogerson and Parry [22] utilized semi-structured interviews to investigate how blockchain works, how it has been developed to enhance visibility and trust along the supply chain, and what limitations it may have. Interestingly, they found that the most appropriate product to adopt blockchain technology is the one for which end-consumers are ready to pay the premium for blockchain, such as baby food. This implies the importance of understanding consumer preferences for adopting a new technology. Jo and Lee [23] argued that the rapid spread of technology has caused the current focus of supply chain management to move from a product-oriented era to a consumer-oriented era. Specifically, in the food industry, they posit that it is essential to identify consumer needs and establish strategies to maximize consumer satisfaction.

Numerous studies have examined consumer preferences and the willingness to pay (WTP) for beef traceability. Dickinson et al. [6] investigated the WTP for red meat traceability in the U.S. and Canada. Consumers in both countries were willing to pay more for a roast beef sandwich with traceability; US $0.21 (7% premium from average bids) for the U.S. and CA $0.45 (9% premium from average bids) for Canada. Further, Americans and Canadians would like to pay even more if the traceability of the roast beef sandwich was bundled with other characteristics, such as animal welfare or enhanced food safety. Dickinson and Bailey [7] conducted follow-up research in the U.K. and Japan. Van et al. [8] utilized means-end-chain laddering to investigate consumers’ perception on traceability. Consumers in the four European countries tend to consider benefits such as health, quality, safety, and control, which are associated with traceability, and traceability may contribute to improving consumer confidence in the product. Loureiro and Umberger [9] conducted choice experiments to investigate U.S. consumers’ relative preferences and the WTP for a ribeye beef steak for attributes like country of origin, traceability, food safety, and tenderness. The WTP for the USDA food safety inspection label is the largest ($8.068/pound), followed by the country-of-origin label ($2.568/pound), the label guaranteeing traceability to the farm ($1.899/pound), and the label verifying guaranteed tender ($0.953/pound). The authors indicated that in cases where the country of origin is associated with a higher food safety perception, it may become a valued quality signal. This research showed that consumers are willing to pay for beef with traceability certification or origin label and tend to associate them with safety or quality perception.

Shew et al. [10] investigated consumers’ preferences for beef-tracking information provided by blockchain technology, and the certification of beef in the U.S.. Intriguingly, they found that respondents did not place much additional value on blockchain-certified beef products compared to USDA-certified products. The authors provided two possible reasons. The first is the respondents’ low familiarity with the blockchain technology. The second explanation is that consumers might care about what blockchain enables, not the technology itself. Lin et al. [11] evaluated Chinese preferences for beef products imported from the U.S. which adopted a blockchain-based traceability system. They estimated that Chinese consumers are willing to pay $0.60/lb more for blockchain-traceable beef compared to beef with the regular digital traceability system. In addition, they conducted a cluster analysis and found that 37% of the respondents belonged to the cluster labeled “Blockchain Enthusiasts”, who’s average WTP for blockchain-traced US beef was about $1.39/lb. This implies that the U.S. beef with blockchain-enabled traceability may potentially be competitive in the Chinese beef market. Differences in the study designs prevent a generalization that Chinese consumers place an intrinsic value on blockchain-based traceability system (Lin et al. [11]), while U.S. consumers do not (Shew et al. [10]); a confound that this study aims to resolve.

Might consumers have an intrinsic value for blockchain technology above and beyond the instrumental value of providing traceability? This research seeks to answer this question by surveying consumers in Hong Kong, South Korea, and the United States. First, we estimate the relative importance of 13 food values, including traceability, based on the best-worst analysis. Second, this research focuses on consumers’ preferences for each type of beef traceability information provided by a blockchain-based system. While previous research investigated consumers’ preferences for beef tracking information provided by blockchain technology, it did not consider the various types of beef tracking information [10,11]. In this research, discrete choice experiments are used to estimate consumers’ preferences and the willingness to pay (WTP) for beef traceability information related to the supply chain, temperature history, and country of origin. Then, we determine how the WTP for the three types of information varied when the information is known to have been provided by the blockchain technology. In the next section of this paper, we will discuss our survey and research methods designed to investigate consumers’ preferences for beef with different types of traceability information. The following section will present the results, and the last section will provide a discussion and conclusions.

## 2. Materials and Methods

### 2.1. Survey

An online survey was conducted in September 2020 where we obtained 1579 completed responses from consumers in three countries; 521 in Hong Kong, 535 in South Korea, and 523 in the U.S. (In 2020, South Korea imports more than 50% of its total beef (chilled/frozen) from the United States, with about 60% of South Korean beef exports going to Hong Kong). Data were acquired from the respondents who are members of a panel maintained by Gallup. The questionnaire was translated into the languages most spoken in each country. Table 1 shows the characteristics of the respondents by country. While the demographic characteristics should be compared to census-level information to determine representativeness, it is of greater interest whether the sample is representative of beef steak consumers. Unfortunately, there is no census-level data on the characteristics of beef consumers. However, to ensure the representativeness of beef steak consumers of this sample, we excluded from the sample the respondents who answered that they do not eat beef. Also, the respondents were asked what percentage of the grocery shopping they do for their household. Overall, 77.6% of respondents answered they do their grocery shopping (73.5% in Hong Kong, 69.5% in South Korea, and 89.9% in the U.S.) indicating our sample could be considered to represent general grocery shoppers.

The survey consists of three parts: introductory questions, choice experiment (CE) questions, and demographic questions. In the introductory part, the survey included questions in which respondents are presented with a trade-off question related to “food values” (Lusk and Briggeman [24]). In particular, the respondents were asked, “How important are the following items to you when deciding whether to buy beefsteak”? Thirteen items were shown, and respondents had to pick four items and drag and drop them into a box labeled “most important” and other items into a box labeled “least important”. This is not a full “best-worst” approach, but represents a faster way to obtain information on the relative importance of attributes in a way that requires respondents to make tradeoffs.

The “food values” questions are analyzed as follows. The 13 items are placed on a relative importance scale ranging from −100% to +100%. The relative importance is calculated as the frequency with which an item was placed in the most important category minus the frequency with which the same item was placed in the least important category. If all respondents placed an issue in the most important category, the score for the issue would be +100%; by contrast if all respondents placed an issue in the least important category, the score for the issue would be −100%. A score of zero would imply that no one put an item in the most or least important categories or that equal frequencies of respondents put an item in the most important category as did the frequency of respondents putting an item in the least important category.

The remaining questions in the survey represent simple Likert-scale questions or straightforward questions about behaviors. Overall, the questions follow advice on good survey questioning and design proposed by authors such as Dillman, Smyth, and Christian [25].

To estimate consumer demand and the WTP for beef steak characteristics, a choice experiment (CE) was created, which required each participant to make repeated choices between different alternatives of beef steak with different attributes and a “none of these” option. The CE method was developed out of the conjoint analysis literature, with a focus on utilizing questioning frameworks consistent with economic theory and were more similar to the sorts of decisions consumers make when actually shopping (Louviere, Hensher, and Swait [26] for compressive treatment of the method). CEs are a popular tool to estimate the value of market and nonmarket goods (Adamowicz et al. [27]; Jayne et al. [28]; Lusk, Roosen, and Fox [29]; Zhang and Sohngen [30]). The method has been shown to exhibit external validity in predicting market behavior in food retail environments (Swait and Andrews [31]; Chang, Lusk, and Norwood [32]; Brooks and Lusk [33]).

To identify the attributes of the beef steak of interest given the study objectives, seven attributes were used in the study design (Table 2). To determine the range of beef steak prices to utilize in the CE, price data from the US Bureau of Labor Statistics was consulted, which indicated that the average U.S. city retail price for USDA Choice beef sirloin steaks ranged from about $8/lb to about $10.50/lb in the two years prior to the survey design. Given that some of the steak attributes are likely to be highly (and lowly) valued by consumers, and the range of the estimated WTP values should fall within the price range used in the design, the experiment design considered prices ranging from $5 to $15 by an increment of $1. To incorporate different levels of average price of beef steak in each country, we added a Hong Kong/South Korea premium to each of the U.S. price levels. That is, the price range of beef in Hong Kong would be from ‘the min price level in the U.S. + (the average beef price level in Hong Kong-the median price level in the U.S.)’ to ‘the max price level in the U.S. + (the average beef price level in Hong Kong-the median price level in the U.S.)’, which is from $13 to $23. We used the same way to calculate the price range of beef steak in South Korea, which is from $28.3 to $38.3 (We used the annual average exchange rate (24 August 2019–24 August 2020) to convert U.S. dollars to Korean won or HK dollars, and vice versa. Also, the average beef steak price level in South Korea and Hong Kong are based on famous online grocery markets, E-Mart Mall and Welcome Supermarkets, respectively. The average price level in South Korea and Hong Kong was 39,350 won ($32.8/lb) and 135 HK dollars ($17.4/lb), respectively, on 24 August 2020). The U.S., Australia, Canada, and Brazil were chosen for the locations of origin because they represent major beef exporting countries.

Even if the prices were varied at only two levels, there are 2^5^ × 3 × 5 = 480 different types of steaks that could be constructed based on the variations in the attributes. To reduce the possibilities, a Bayesian optimal experimental design was constructed to minimize the standard errors of a multinomial logit choice model (i.e., to extract as much information as possible about consumer preferences while only asking consumers a reasonable small number of choices), e.g., see Kessels et al. [34] or Scarpa et al. [35]. The standard errors of the multinomial logit model depend on the true underlying parameters, and it is in this sense that the design is Bayesian: priors for the preference parameters are needed to calculate D-efficiency. In each choice question, we chose to present respondents with three beef steak alternatives and a “none of these” option. We chose a design that required the use of 20 total choice questions, and because 20 choice questions are probably too many for each person to answer, the choice questions were blocked into sets of 10 questions, and each respondent should be randomly assigned to one of the blocks. Several constraints were placed on the optimization of the D-efficiency criteria. In particular, options which provided no information about traceability to the farm, temperature, or origin could not also provide the information about blockchain, and as such, these alternatives were eliminated, as were options that indicated the option was traceability to the farm, but country of origin was unknown. The design was created using the software NGENE 1.3 (Ngene code we used is in the Appendix B). The software searches through all possible combinations of choice options to find the design that maximizes the D-efficiency criteria given the aforementioned constraints (This would yield a sampling error of +/−4.4% on a single binary choice question. However, note that for the choice experiment, each respondent answered 10 choice questions, which would yield a sampling error of about 1.4%. The Choice Experiment was developed using well-established methodologies, and the approach has shown high level of external validity (e.g., Lusk and Brooks, 2010; Swait and Andrews)).

Figure 1 shows an example of choice questions used in this research. In each choice question, we chose to present respondents with three beef steak alternatives and a “none of these” option. (The questionnaires for this survey are provided as Appendix A). We chose a design that required the use of 20 total choice questions, and because 20 choice questions are probably too many for each person to answer without a degrading response quality, the choice questions were blocked into sets of 10 questions, and each respondent was randomly assigned to one of the two blocks. To assess the effects of technology information on consumer preferences, we provided the information statement to only one of the blocks—treatment 1 (control, no information statement provided) and treatment 2 (the provision of the information statement).

Figure 2 and Figure 3 below are the statement we used in the survey. The Figure 2 statement is for the respondents where no information about the blockchain technology is provided, and the respondents who are randomly assigned to the block which provides the information about the blockchain technology saw the Figure 3 statement. The information statement is about the concept of the blockchain technology, which is supposed to be utilized to track the location during all parts of the supply chain back to the farm, temperature, and origin information in the experiment.

### 2.2. Econometric Analysis

Consumer *i* in country *t* is assumed to derive the following utility from choice option *j*: $U_{itj}=V_{tj}+\varepsilon_{itj}$. If the $\varepsilon_{itj}$ follow a Type I extreme value distribution and are independently and identically distributed across *i*, *t*, and *j*, then the conventional multinomial logit model (MNL) results:(1)$$\mathrm{Prob}\;(i\;\mathrm{chooses}\;j\;\mathrm{in}\;\mathrm{country}\;t)=\frac{e^{V_{tj}}}{\sum_{k=1}^{3}e^{V_{tk}}}$$

The systematic portion of the utility function is posited to be a linear function of steak attributes:(2)$$\begin{array}{cc} V_{itj}=\beta_{it}{None}_{jt} & +\alpha_{it}p_{j}+\theta_{it}^{1}{SCTr}_{j}+\theta_{it}^{2}{SCTr\_BC}_{j}+\theta_{it}^{3}{temp\_pos}_{j}\\ & +\theta_{it}^{4}{temp\_neg}_{j}+\theta_{it}^{5}{temp\_BC}_{j}+\theta_{it}^{6}{origin\_US}_{j}\\ & +\theta_{it}^{7}{origin\_SK}_{j}+\theta_{it}^{8}{origin\_AUS}_{j}+\theta_{it}^{9}{origin\_CAN}_{j}\\ & +\theta_{it}^{10}{origin\_BRZ}_{j}+\theta_{it}^{11}{origin\_BC}_{j} \end{array}$$
where

$p_{j}$ is the price of alternative j,

$\alpha_{t}$ is the marginal utility of a price change in location *t*,

$\beta_{it}$ is the utility of the “no purchase” option relative to the utility of one of the steak alternatives,

$None_{jt}$ = 1 if the option is the “none of these” option; 0 otherwise

$SCTr_{j}$ = 1 if option j is supply chain traceable; 0 otherwise,

$SCTr\_BC_{j}$ = 1 if supply chain traceability is provided by blockchain; 0 otherwise,

$temp\_{\mathrm{pos}}_{j}$ = 1 if temperature history is available and indicates it never exceeded save levels; 0 otherwise,

$temp\_{\mathrm{neg}}_{j}$ = 1 if temperature history is available and indicates it exceeded safe levels for 20 min; 0 otherwise,

$temp\_BC_{j}$ = 1 if temperature history is provided by blockchain; 0 otherwise,

$origin\_US_{j}$ = 1 if country of origin is United States; 0 otherwise,

$origin\_SK_{j}$ = 1 if country of origin is South Korea; 0 otherwise,

$origin\_AUS_{j}$ = 1 if country of origin is Australia; 0 otherwise,

$origin\_CAN_{j}$ = 1 if country of origin is Canada; 0 otherwise,

$origin\_BRZ_{j}$ = 1 if country of origin is Brazil; 0 otherwise,

$origin\_BC_{j}$ = 1 if country of origin information is provided by blockchain; 0 otherwise,

$\theta_{it}^{1}$ is the utility of supply chain traceability relative to no supply chain traceability,

$\theta_{it}^{2}$ is the utility of supply chain traceability provided by blockchain relative to the method being unspecified,

$\theta_{it}^{3}$ is the utility of temperature history is available and indicates it never exceeded save levels relative to no temperature history being available,

$\theta_{it}^{4}$ is the utility of temperature history is available and indicates exceeded save levels for 20 min relative to no temperature history being available,

$\theta_{it}^{5}$ is the utility of temperature history being provided by blockchain relative to the method being unspecified,

$\theta_{it}^{6}$ is the utility of the U.S. beef relative to origin being unknown,

$\theta_{it}^{7}$ is the utility of South Korean beef relative to origin being unknown,

$\theta_{it}^{8}$ is the utility of Australian beef relative to origin being unknown,

$\theta_{it}^{9}$ is the utility of Canadian beef relative to origin being unknown,

$\theta_{it}^{10}$ is the utility of Brazilian beef relative to origin being unknown,

$\theta_{it}^{11}$ is the utility of country of origin information being provided by blockchain relative to the method being unspecified.

A key downside to the MNL is that it assumes all consumers have the same preference. Moreover, the MNL imposes the potentially restricted independence of the irrelevant alternatives assumption. Thus, we estimate a random parameter logit (RPL) model that relaxes these restrictive assumptions (Train [36]). The coefficients are specified as, $\beta_{ik}={\bar{\beta }}_{k}+\sigma_{k}\lambda_{ik}$, where ${\bar{\beta }}_{k}$ is the mean preference parameter for attribute *k*, $\sigma_{k}$ is the standard deviation of preferences, and $\lambda_{ik}$ is a random term that distributed Normal, N(0,1). To ensure consistency with economic theory, and to constrain consumers to have a negative marginal utility of price increases, the price coefficient was assumed non-random. We estimated the RPL parameters via simulation using 500 individual-specific Halton draws (Train [36]).

Based on the estimates, the willingness-to-pay (WTP) can be estimated as the dollar premium that would induce a consumer to be exactly indifferent to buying a steak option with one set of characteristics vs. another steak option (or “none”) with a different set of characteristics. For example, the maximum premium a consumer would be willing to pay to have a steak with supply chain traceability vs. one that is not supply chain traceable is $-\theta_{it}^{1}/\alpha_{t}$. The maximum premium a consumer would be willing to pay to have a steak with supply chain traceability provided by blockchain vs. one that is not supply chain traceable is $-(\theta_{it}^{1}+\theta_{it}^{2})/\alpha_{t}$.

## 3. Results

Respondents were asked “How important are the following items to you when deciding whether to buy beef steak”? Table 3 shows the relative importance of the 13 items, ranging from −100% to +100%, while Figure 4 summarizes their rank by country. The most important food values when purchasing beef steak across for all three countries included price, taste, and safety. Fairness and novelty are least important in all three countries. As such, there are food values that are the most important or the least important that the countries surveyed shared in common, while others have different relative importance depending on the country.

Interestingly, while the Asian countries, South Korea, and Hong Kong, show similar patterns to each other, they present different patterns from those of the U.S. which belongs to the Western world. In the U.S., appearance is considered as the fourth most important value after price, taste, and safety. However, in South Korea and Hong Kong, the percentage of putting appearance in the least important category is larger than that in the most important category. Appearance in South Korea and Hong Kong ranks the eighth and the seventh, respectively, which is relatively lower than that in the U.S.. Animal welfare shows similar patterns with appearance. On the other hand, there are some values which are located in the higher rank in South Korea and Hong Kong compared to the U.S.; origin, convenience, and environmental impact are included in this case. In particular, while origin is the third and the fourth most important value in South Korea and Hong Kong, respectively, there are more people who placed origin in the least important category in the U.S.

We estimated an RPL model by each country and conducted a likelihood ratio test to determine if the estimates vary by country. We compared the sum of the likelihood function values from each country to the pooled model and found a chi-square value of 1549.9 with 26 degrees of freedom. The 95% critical chi square value is only 38.89, and therefore we reject the null hypothesis that preferences are equal across countries. Thus, we report results separately for each country.

Table 4 reports the RPL estimates for each treatment in Hong Kong. We conducted a likelihood ratio test of the null hypothesis that the coefficients are equal across treatments. The null hypothesis is rejected at the *p* < 0.01 level, indicating that the provision of information significantly affected the parameter estimates—a chi-square of 29.6 with 13 degrees of freedom. The sign of price coefficient is negative in both treatments, indicating that the utility decreases with a price increase. This result accords with the ordinary demand curve. The coefficient of none is negative, meaning the respondents prefer purchasing one of the suggested beef options rather than buying nothing. In the control treatment, consumers prefer beef with supply chain traceability (SCTr), traceability of temperature history (Temp_Pos), and traceability of country-of-origin information (Origin_US, Origin_SK, Origin_AUS, Origin_CAN, Origin_BRZ) compared to beef with no information available. However, they do not significantly care whether this information was provided by blockchain technology or not. The signs of SCTr_BC, and Temp_BC are both positive, but not statistically significant from zero. Even the sign of Origin BC is negative, indicating that consumers do not prefer beef with origin information utilizing blockchain to the one with not utilizing blockchain.

Interestingly, in treatment 2, in which the information statement regarding the concept of the blockchain technology was provided, consumers prefer the supply chain traceability information using blockchain more than the one without blockchain-enabled traceability. Further, we find that the sign of coefficients of Temp_Neg was positive and statistically significant in treatment 2, meaning consumers prefer purchasing beef with negative temperature history information (temperature exceeding the safe level for 20 min) to the one with no temperature information, perhaps as a result of uncertainty or ambiguity aversion.

Table 5 shows the RPL estimates for South Korea. We conducted a likelihood ratio test to determine whether the coefficients are statistically different by treatment. The results show a chi-square value of 53 with 13 degrees of freedom, which can be contrasted with the 95% critical chi-square value of 22.4, indicating that we can reject the null hypothesis of equal preferences in both information treatments. The price coefficient presents statistically significant negative signs in both treatments. Also, we could find the negative sign for the none coefficient in both treatments, meaning that consumers would like to buy one of the options rather than nothing at all.

In both treatments, the respondents preferred to purchase the beef with supply chain traceability, temperature history, and country of origin information rather than the one without any information available. In particular, South Korean consumers prefer the beef if the country-of-origin information is provided by the blockchain technology compared to not using the technology. The sign of Origin_BC is positive and statistically significant in both treatments. As is shown in Table 3, the rank of the relative importance of the origin value in South Korea is relatively higher than those in the U.S., and Hong Kong, which matches up to the RPL model result. It manifests the level of importance that Korean consumers put on the country of origin information when buying beef. What is more interesting is that the sign of Temp_Neg parameter has changed from negative in treatment 1 to positive in treatment 2. This indicates that consumers tend to purchase beef even with negative temperature history information (temperature exceeding the safe level for 20 min) rather than the one with no temperature history information available in the group for which the information regarding blockchain was provided.

Table 6 shows the estimates of the RPL model in the U.S. Like in the two previous cases, we conducted a likelihood ratio test of the null hypothesis that the coefficients are equal across treatments. It yields a chi-square value of 122.4 (−2 × (Total-sum (T1; T2)) with 13 degrees of freedom, indicating that the null hypothesis is rejected at the *p* < 0.01 level. The sign of the price coefficient is negative and statistically significant, indicating that consumer utility increases with a price decrease. The coefficient of the opt-out option was in the positive, which means that consumers prefer an opt-out option relative to one of the beef products. However, the standard deviation for the none parameter was large and statistically significant in both treatments, implying there is significant preference heterogeneity in the population. Also, the coefficient of the none is not statistically different from zero in treatment 2.

In the U.S., consumers prefer tracking temperature history information with blockchain compared to not tracking with blockchain. However, they did not like to have the country-of-origin information provided by the blockchain technology in both treatments. The sign of SCTr_BC has turned to positive in treatment 2, but it is not still statistically significant. In summary, different countries have different preferences for the type of information provided by the blockchain technology—supply chain traceability information for Hong Kong, the country-of-origin information for South Korea, and temperature history information for the U.S.

The models in Table 4, Table 5 and Table 6 can be used to determine the consumer WTP for supply chain traceability, temperature history traceability, and country of origin in beef steaks in the presence or absence of information about blockchain technology. Table 7 and Table 8 report the WTP for each type of information vs. no information available (A) and the WTP for each information provided by blockchain vs. no information available (B). Since we used different price levels for each country, the WTP range differs by country; between $13~$23 for Hong Kong, between $28.3~$38.3 for South Korea, and between $5~$15 for the U.S.

Looking at the result by the type of information, for the temperature history, all three countries demonstrate an increased WTP when the temperature information is provided by blockchain in all the treatments. This indicates that the temperature history information depends relatively highly on the way the information is provided. Furthermore, in Hong Kong and the U.S., providing the information on the technology (T2) raises the differences between A and B—from $0.06 to $0.12 in Hong Kong and from $1.57 to $3.21 in the U.S., meaning that enhancing the consumers’ knowledge level on the technology utilized would positively influence their decision on beef purchases. Interestingly, the premium that consumers are willing to pay for beef with even negative temperature information (temperature exceeding the safe level for 20 min) is larger than the one with no temperature information available when temperature information is provided by blockchain. This implies that the blockchain technology reinforces the consumers’ trust on temperature traceability information, thus they would like to pay more for the information provided by blockchain.

Unlike the temperature information, for the country-of-origin information, the countries showed different tendencies. As shown in Table 7 and Table 8, it was only South Korea that had higher ‘WTP for origin information provided by blockchain vs. no information available’ (B). That is, it might be worth applying the blockchain technology to track the country of origin in the South Korean beef market, but not in Hong Kong and the U.S. Moreover, Korean consumers have the highest WTP for domestic beef ($28.14/lb (A), $30.82/lb (B) in T1 and $20.99/lb (A), $23.04/lb (B) in T2) followed by Australian beef. The Korean consumers’ higher preferences on domestic beef compared to the imported beef was also shown in the study by Chung et al. [37]. In the study they conducted in South Korea, they found that consumers valued $14/lb more for domestic beef than the imported beef.

As for the supply chain traceability information, Hong Kong and Korean consumers turned out to have an increased WTP for beef with supply chain traceability when it is provided by the blockchain technology in the control (T1) and information treatment (T2). In other words, utilizing the blockchain technology to track the supply chain traceability in beef steak would strengthen the consumers’ trust on the information and lead to a higher WTP for the product. Intriguingly, consumers in the U.S. showed a larger WTP for beef with supply chain information provided by blockchain (B) rather than that for supply chain information (A) in the information treatment (T2), which was not the case in the control treatment (T1), with the increment ($3.51/lb) being the largest among the three countries.

## 4. Discussion

Foodborne illnesses and vulnerabilities in current food traceability systems, such as the inability to link food chain records, inaccuracy and errors in records, and delays in obtaining essential data, have emerged as serious issues (WHO [1]; Badia-Melis et al. [2]). As previously mentioned (Bischoff and Seuring [4]), the immutability of blockchain, which ensures tamper-proof data sharing, is key to resolving the longstanding problems of food fraud and quality. With this trend, blockchain applications in food traceability are rapidly growing (Market and Market [5]). This study suggests blockchain technology as a solution to the current issues and analyzes consumers’ preferences for its effective implementation.

In this research, we conducted a survey in three different countries to investigate consumers’ preferences for beef traceability information and the technology for conveying the information. Interestingly, each country demonstrated different preferences for beef traceability information. Overall, consumers in the three countries tend to prefer beef with traceability information—supply chain traceability, temperature history, and country of origin. However, the intrinsic value of using blockchain to deliver the information differed by country and by attribute. For example, consumers in Hong Kong prefer beef in which supply chain traceability information is provided by blockchain, in Korea, country-of-origin information, and in the U.S., temperature history information. When blockchain is applied to the food industry, one of the important considerations is which type of information would be handled by blockchain. This is because the cost cannot be ignored. Thus, the previous research we reviewed also describes in detail which data is tracked in the blockchain (Feng et al. [12]; Iftekhar and Xiaohui [13]; Bumblauskas et al. [14]). The result of this study suggests that it is not necessary to provide all types of traceability information of beef with blockchain. It might be more efficient to provide only the type of information with blockchain for which consumers are willing to pay.

In November 2018, the Korean government implemented a pilot project to apply blockchain to ‘Animal Products Traceability’ in a region (North Jeolla Province) for a year. This was to prevent forgery and falsification of various documents required for the Animal Products Traceability, and to shorten the traceability time from 5 days to 10 min in the event of a livestock disease. The result of this project confirmed that it is technically possible to introduce the blockchain technology. However, a few factors to be overcome that were pointed out in order to fully commercialize the system included the large amount of information to be managed by the blockchain platform and the cost of equipment (e.g., IoT sensors) necessary to manage the information with the blockchain platform. The research results reviewed in the previous paragraph may be utilized as one of the possible solutions to this problem. In the early stages when the blockchain platform is introduced, we need to be selective about the type of information to apply the technology to due to cost. One way to go about this would be to analyze which information the consumers in each country are willing to pay for a blockchain platform, and then gradually expand the platform from there. Furthermore, this implication holds valuable information for food product producers, manufacturers, and even retailers seeking to meet consumer satisfaction regarding food safety and supply chain transparency. With the use of our estimates of the WTP, they could effectively develop a cost-efficient blockchain based food cold chain system.

In all the three countries, consumers show positive preferences for the negative temperature history of beef relative to no temperature information being available. While somewhat surprising, the behavior is consistent with a high degree of uncertainty and ambiguity aversion. This implies that the consumers’ demand is growing stronger for the food cold chain management information and resolving the information asymmetry. In other words, it would be desirable for governments and stakeholders in the food industry to establish a strategy to secure the consumers’ trust in the product by sharing all collected information, including negative information, with the consumers, and not by only emphasizing the positive aspects of the product.

We compared the WTP for each type of traceability information of beef and found two intriguing issues. The first is that only Korean consumers have an intrinsic value for blockchain in providing the information of country of origin. In practice, as we can see in food value analysis, origin value is the third most important value in South Korea. The U.S. shows the largest increase in the WTP over all types of information provided by blockchain in the information treatment (T2). This indicates that the provision of information regarding the concept of the blockchain technology may have more impact on the U.S. consumers compared to the consumers in Hong Kong and South Korea. In other words, the consumers’ prior knowledge about a new technology or system which is applied to food products might have a larger impact on their actual food purchase in the U.S. than in Hong Kong and South Korea. This leaves an important implication for the U.S. government to establish a strategy to advance the food cold chain. Introducing an advanced food cold chain system must come after the national consensus is reached on the necessity of the system. What can be learned from this study is that reaching an effective national consensus requires that the public awareness on the system itself should be raised before the need for the system is directly mentioned.

## 5. Conclusions

In this research, we tried to provide practical implications to the government and relevant market officials who want to introduce the blockchain technology to beef traceability systems by identifying the intrinsic and instrumental value of blockchain to provide beef traceability in three countries. We could see from the results of this study that consumers do not think that all information should be managed with the blockchain technology. They also want to minimize information asymmetry and are looking for reliable information. This study is meaningful in that it proposes the most cost-effective approach while satisfying these consumer needs.

Despite the plethora of findings in this research, more remains to be learned. In this research, the information provided to respondents is a neutral definition; however, it would be interesting to explore how the nature of delivering the information on the blockchain technology (favorable or critical) may influence the consumers’ preference. This exploration holds particular value for government and stakeholders in the food supply chain aiming to maximize the impact of this system on policy and sales. Additionally, a within-subject-design could provide insights on how prior knowledge and prior preferences might influence information assimilation. Lastly, research on the data standardization and strategies for building consensus among supply chain stakeholders for blockchain based digital system is needed. Discrete choice experiments might be a useful method to estimate the incentives for stakeholders, encouraging their participation in the system.

## Figures and Tables

**Figure 1 foods-12-04209-f001:**
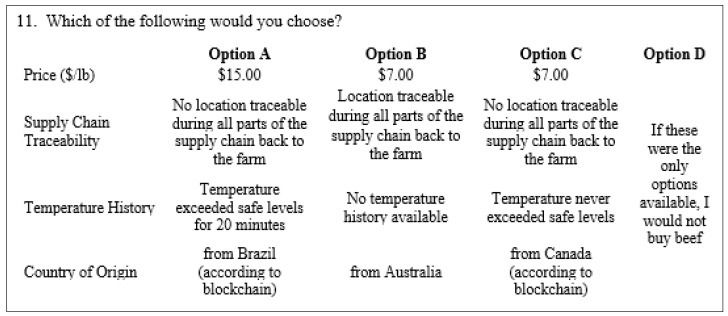
The example of choice question.

**Figure 2 foods-12-04209-f002:**
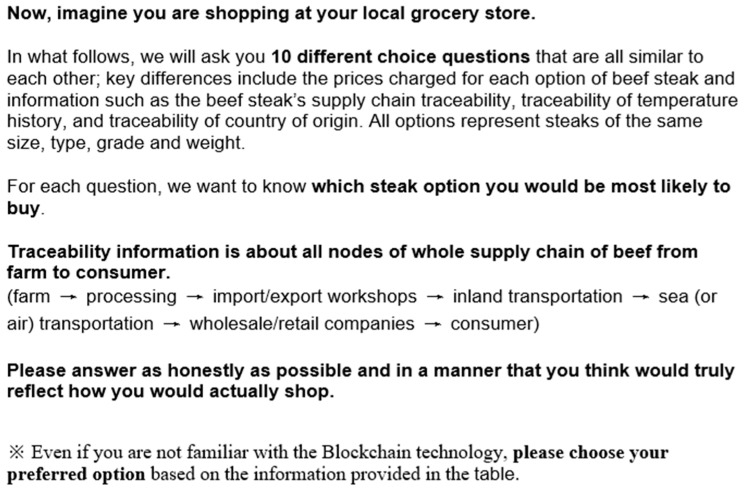
The statement provided to respondents in the absence of information about blockchain technology.

**Figure 3 foods-12-04209-f003:**
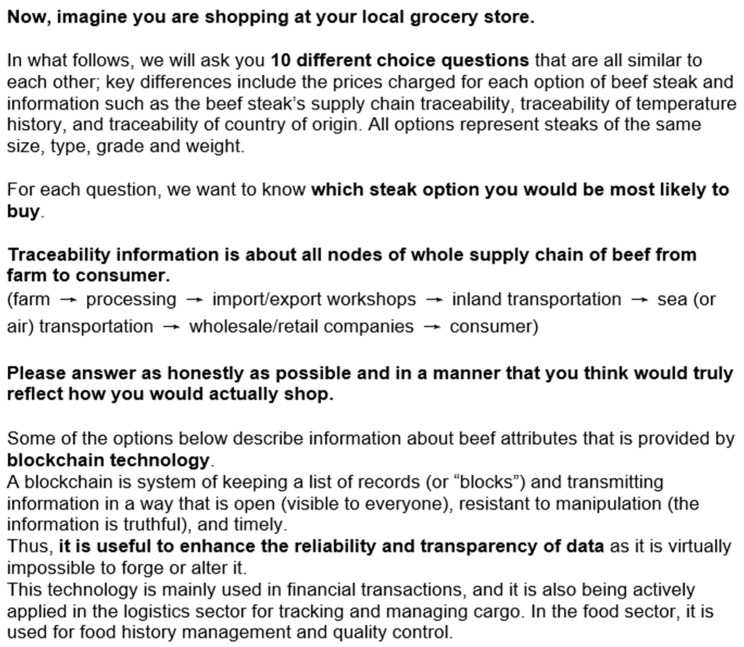
The statement provided to respondents when information about blockchain technology is available.

**Figure 4 foods-12-04209-f004:**
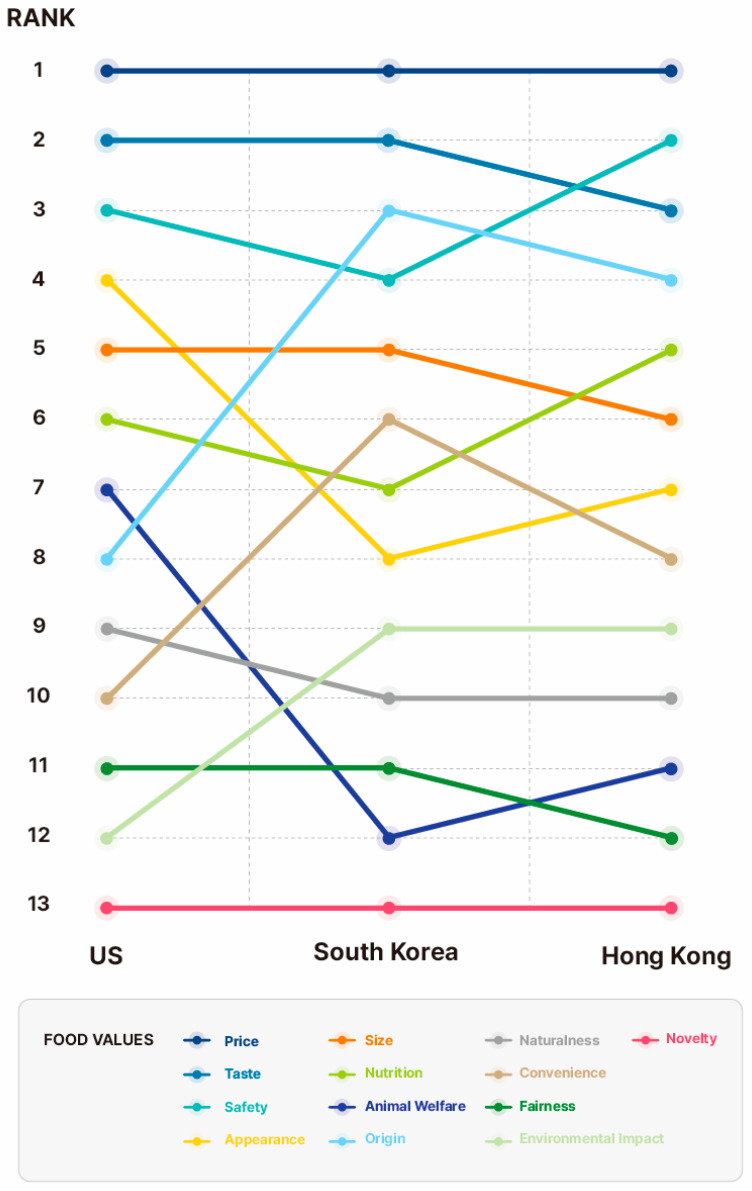
The rank of relative importance of 13 food values by country.

**Table 1 foods-12-04209-t001:** Socio-demographic characteristics of the sample (%).

Characteristics		Hong Kong (N = 521)	South Korea (N = 535)	U.S. (N = 523)	Total (N = 1579)
Gender	Male	50.1	51.0	49.9	50.3
	Female	49.9	49.0	50.1	49.7
Age	20~29	24.0	22.8	23.5	23.4
	30~39	26.5	26.2	26.4	26.3
	40~49	26.3	26.9	26.0	26.4
	Over 50	23.2	24.1	24.1	23.8
Marital Status	Single	42.0	43.2	37.1	40.8
	Married	51.1	53.8	45.1	50.0
	Separated	3.1	0.2	3.4	2.2
	Divorced	3.1	2.4	12.2	5.9
	Widowed	0.8	0.4	2.1	1.1
Education level	Less than high school	3.8	0.2	3.6	2.5
	High school/GED	20.9	17.8	26.0	21.5
	Some college	10.2	2.4	25.8	12.7
	2-Year college degree	9.2	16.1	10.5	12.0
	4-Year college degree	40.5	52.9	21.2	38.3
	Master’s degree	13.4	9.0	11.9	11.4
	Professional degree	1.9	1.7	1.0	1.5
Primary shopper	100%	21.3	25.8	53.9	33.6
	75~99%	25.1	21.5	21.8	22.8
	50~74%	27.1	22.2	14.1	21.2
	25~49%	18.8	19.8	6.1	14.9
	1~25%	6.7	9.5	2.7	6.3
	0%	1.0	1.1	1.3	1.1

**Table 2 foods-12-04209-t002:** Attributes and level.

Attributes	Level
Price	Hong Kong: varies between HK$100/lb and HK$178/lb South Korea: varies between 33,900 won/lb and 45,900 won/lb the U.S.: varies between $5/lb and $15/lb
Supply chain traceability ^1^	Yes/No
Whether supply chain traceability was provided by blockchain	Yes/No
Temperature history	no temperature information available/temperature never exceeded safe levels/temperature exceeded safe levels for 20 min
Whether temperature was tracked by blockchain	Yes/No
Country of origin	Unknown/United States/South Korea/Australia/Canada/Brazil
Whether country of origin was tracked by blockchain	Yes/No

^1^ Supply chain traceability implies that no location traceable during all parts of the supply chain back to the farm.

**Table 3 foods-12-04209-t003:** Relative importance of 13 food values by country.

Food Values	U.S.	South Korea	Hong Kong
Price	52.6%	70.7% (1) *	53.9% (1)
Taste	40.3%	62.8% (2)	31.9% (3)
Safety	31.2%	29.5% (4)	33.6% (2)
Appearance	22.2%	−20.4% (8)	−3.6% (7)
Size	18.0%	15.7% (5)	2.3% (6)
Nutrition	−2.7%	−17.9% (7)	6.0% (5)
Animal Welfare	−7.1%	−41.3% (12)	−24.2% (11)
Origin	−11.3%	52.0% (3)	7.9% (4)
Naturalness	−19.3%	−28.2% (10)	−16.9% (10)
Convenience	−22.8%	−5.0% (6)	−8.6% (8)
Fairness	−22.9%	−32.9% (11)	−26.5% (12)
Environmental Impact	−24.5%	−27.3% (9)	−16.3% (9)
Novelty	−53.7%	−57.6% (13)	−39.3% (13)

Note: Food values are in descending order based on the U.S. values. * Parentheses indicate the ranking of food values.

**Table 4 foods-12-04209-t004:** Estimates of RPL model in Hong Kong.

Parameters	Control Treatment (T1)	Information Treatment (T2)	Pooled
SCTr	0.425 ***	0.552 ***	0.459 ***
	(0.076)	(0.085) a	(0.057)
	[0.679] ***	[0.798] ***	[0.770] ***
	(0.100)	(0.117) b	(0.066)
SCTr_BC	0.018	0.150 *	0.058
	(0.077)	(0.079)	(0.053)
	[0.446] ***	[0.528] ***	[0.334] ***
	(0.137)	(0.128)	(0.097)
Temp_Pos	0.614 ***	0.955 ***	0.749 ***
	(0.105)	(0.114)	(0.075)
	[1.019] ***	[1.253] ***	[1.070] ***
	(0.094)	(0.107)	(0.078)
Temp_Neg	0.099	0.202 **	0.135 **
	(0.084)	(0.100)	(0.066)
	[0.320] **	[0.621] ***	[0.554] ***
	(0.143)	(0.125)	(0.116)
Temp_BC	0.006	0.015	0.008
	(0.059)	(0.062)	(0.042)
	[0.041]	[0.085]	[0.0001]
	(0.118)	(0.133)	(0.090)
Origin_US	1.640 ***	1.897 ***	1.751 ***
	(0.198)	(0.221)	(0.146)
	[0.248]	[0.844] ***	[0.509] ***
	(0.296)	(0.122)	(0.118)
Origin_SK	1.492 ***	1.929 ***	1.653 ***
	(0.201)	(0.215)	(0.146)
	[0.883] ***	[0.896] ***	[0.971] ***
	(0.119)	(0.124)	(0.086)
Origin_AUS	2.017 ***	2.406 ***	2.105 ***
	(0.200)	(0.221)	(0.145)
	[0.508] ***	[0.838] ***	[0.587] ***
	(0.121)	(0.098)	(0.085)
Origin_CAN	1.849 ***	2.143 ***	1.975 ***
	(0.201)	(0.215)	(0.144)
	[0.416] **	[0.469] ***	[0.282] **
	(0.196)	(0.122)	(0.140)
Origin_BRZ	0.995 ***	1.335 ***	1.118 ***
	(0.200)	(0.206)	(0.143)
	[0.821] ***	[0.597] ***	[0.761] ***
	(0.155)	(0.167)	(0.126)
Origin_BC	−0.027	−0.003	−0.013
	(0.056)	(0.061)	(0.041)
	[0.124]	[0.110]	[0.024]
	(0.129)	(0.105)	(0.078)
None	−2.716 ***	−1.885 ***	−2.425 ***
	(0.381)	(0.329)	(0.249)
	[3.316] ***	[2.892] ***	[3.187] ***
	(0.326)	(0.226)	(0.307)
Price	−0.109 ***	−0.129 ***	−0.118 ***
	(0.013)	(0.014)	(0.009)
N individuals	261	260	521
Log Likelihood Function	−2886.3	−2839.8	−5740.9

Note: Number in parentheses ( ) a are standard errors of mean importance of the value. Number in brackets [ ] are standard deviation. Number in parentheses ( ) b are standard errors of standard deviation of the value. An * denotes significance at the 10% level, ** denotes significance at the 5% level, and *** denotes significance at the 1% level.

**Table 5 foods-12-04209-t005:** Estimates of RPL model in South Korea.

Parameters	Control Treatment (T1)	Information Treatment (T2)	Pooled
SCTr	1.392 ***	1.276 ***	1.277 ***
	(0.119)	(0.115) a	(0.077)
	[1.163] ***	[1.115] ***	[1.043] ***
	(0.143)	(0.132) b	(0.086)
SCTr_BC	0.118	0.139	0.100
	(0.095)	(0.086)	(0.062)
	[0.784] ***	[0.427] ***	[0.578] ***
	(0.169)	(0.147)	(0.114)
Temp_Pos	1.990 ***	1.891 ***	1.964 ***
	(0.143)	(0.137)	(0.099)
	[1.136] ***	[1.415] ***	[1.296] ***
	(0.110)	(0.127)	(0.086)
Temp_Neg	−0.026	0.291 **	0.150 *
	(0.137)	(0.116)	(0.088)
	[0.878] ***	[0.435] **	[0.659] ***
	(0.155)	(0.202)	(0.117)
Temp_BC	0.035	0.009	0.027
	(0.074)	(0.069)	(0.049)
	[0.266] **	[0.059]	[0.110]
	(0.135)	(0.129)	(0.104)
Origin_US	1.988 ***	1.724 ***	1.754 ***
	(0.334)	(0.287)	(0.217)
	[0.156]	[0.501] ***	[0.492] ***
	(0.647)	(0.189)	(0.149)
Origin_SK	3.968 ***	3.379 ***	3.601 ***
	(0.331)	(0.276)	(0.218)
	[1.480] ***	[1.185] ***	[1.248] ***
	(0.162)	(0.157)	(0.092)
Origin_AUS	2.752 ***	2.306 ***	2.434 ***
	(0.327)	(0.274)	(0.205)
	[0.376] *	[0.352] **	[0.186]
	(0.195)	(0.177)	(0.150)
Origin_CAN	2.342 ***	2.079 ***	2.131 ***
	(0.324)	(0.272)	(0.204)
	[0.451] **	[0.392]	[0.378] **
	(0.192)	(0.246)	(0.189)
Origin_BRZ	1.445 ***	1.004 ***	1.198 ***
	(0.323)	(0.278)	(0.204)
	[0.929] ***	[1.246] ***	[0.896] ***
	(0.273)	(0.185)	(0.176)
Origin_BC	0.377 ***	0.331 ***	0.339 ***
	(0.071)	(0.069)	(0.049)
	[0.149]	[0.266] **	[0.218] **
	(0.227)	(0.112)	(0.092)
None	−2.677 ***	−3.887 ***	−3.363 ***
	(0.670)	(0.661)	(0.445)
	[2.648] ***	[3.681] ***	[3.055] ***
	(0.241)	(0.383)	(0.191)
Price	−0.141 ***	−0.161 ***	−0.146 ***
	(0.020)	(0.019)	(0.013)
N individuals	270	265	535
Log Likelihood Function	−2369.3	−2414.2	−4810.0

Note: Number in parentheses ( ) a are standard errors of mean importance of the value. Number in brackets [ ] are standard deviation. Number in parentheses ( ) b are standard errors of standard deviation of the value. An * denotes significance at the 10% level, ** denotes significance at the 5% level and *** denotes significance at the 1% level.

**Table 6 foods-12-04209-t006:** Estimates of RPL model in the U.S.

Parameters	Control Treatment (T1)	Information Treatment (T2)	Pooled
SCTr	0.534 ***	0.458 ***	0.480 ***
	(0.088)	(0.095) a	(0.064)
	[0.510] ***	[0.786] ***	[0.829] ***
	(0.135)	(0.106) b	(0.078)
SCTr_BC	−0.013	0.137	0.044
	(0.090)	(0.091)	(0.062)
	[0.489] ***	[0.386] ***	[0.328] **
	(0.169)	(0.127)	(0.146)
Temp_Pos	1.330 ***	1.055 ***	1.145 ***
	(0.137)	(0.143)	(0.098)
	[1.582] ***	[1.378] ***	[1.424] ***
	(0.129)	(0.116)	(0.102)
Temp_Neg	0.243 **	0.012	0.131
	(0.109)	(0.139)	(0.082)
	[0.699] ***	[1.016] ***	[0.740] ***
	(0.139)	(0.180)	(0.119)
Temp_BC	0.130 *	0.125 *	0.100 **
	(0.073)	(0.075)	(0.049)
	[0.389] ***	[0.116]	[0.029]
	(0.105)	(0.173)	(0.103)
Origin_US	2.390 ***	2.022 ***	2.261 ***
	(0.228)	(0.230)	(0.158)
	[1.583] ***	[1.290] ***	[1.294] ***
	(0.134)	(0.129)	(0.088)
Origin_SK	0.636 ***	0.165	0.421 ***
	(0.220)	(0.225)	(0.154)
	[0.731] ***	[0.923] ***	[0.912] ***
	(0.140)	(0.142)	(0.112)
Origin_AUS	0.958 ***	0.720 ***	0.801 ***
	(0.212)	(0.217)	(0.150)
	[0.494 ***	[0.476] ***	[0.278]
	(0.159)	(0.182)	(0.198)
Origin_CAN	1.302 ***	1.033 ***	1.202 ***
	(0.213)	(0.211)	(0.146)
	[0.734] ***	[0.315] *	[0.041]
	(0.129)	(0.169)	(0.212)
Origin_BRZ	0.586 ***	0.220	0.482 ***
	(0.209)	(0.213)	(0.143)
	[0.287]	[0.575] ***	[0.188]
	(0.207)	(0.159)	(0.246)
Origin_BC	−0.054	−0.055	−0.069
	(0.075)	(0.072)	(0.049)
	[0.494] ***	[0.164]	[0.198] *
	(0.096)	(0.122)	(0.102)
None	0.432 *	0.416	0.826 ***
	(0.239)	(0.285)	(0.180)
	[3.594] ***	[2.958] ***	[2.898] **
	(0.252)	(0.230)	(0.175)
Price	−0.083 ***	−0.039 ***	−0.058 ***
	(0.014)	(0.015)	(0.010)
N individuals	266	257	523
Log Likelihood Function	−2766.4	−2671.3	−5498.9

Note: Number in parentheses ( ) a are standard errors of mean importance of the value. Number in brackets [ ] are standard deviation. Number in parentheses ( ) b are standard errors of standard deviation of the value. An * denotes significance at the 10% level, ** denotes significance at the 5% level and *** denotes significance at the 1% level.

**Table 7 foods-12-04209-t007:** Willingness to pay (WTP) for each type of information provided by blockchain ($/lb) in three different countries under the condition of no information being given.

	Control; No Information Condition (T1)					
	Hong Kong		South Korea		U.S.	
Attribute	Without Block-Chain	With Block-Chain	Without Block-Chain	With Block-Chain	Without Block-Chain	With Block-Chain
SCTr *	3.90	4.06	9.87	10.71	6.43	6.28
Temp_pos **	5.63	5.69	14.11	14.36	16.02	17.59
Temp_neg ***	0.91	0.96	−0.18	0.06	2.93	4.49
USc	15.06	14.81	14.10	16.77	28.80	28.14
SKc	13.69	13.44	28.14	30.82	7.66	7.01
AUSc	18.50	18.26	19.52	22.19	11.54	10.89
CANc	16.96	16.72	16.61	19.28	15.69	15.04
BRZc	9.13	8.88	10.25	12.92	7.06	6.41

* WTP for beef supply chain traceability vs. beef not supply chain traceable. ** WTP expressed relative to beef with temperature history unknown. *** WTP for beef from respective origin relative to origin unknown.

**Table 8 foods-12-04209-t008:** Willingness to pay (WTP) for each type of information provided by blockchain ($/lb) in three different countries under the condition of information being given.

	Information Treatment (T2)					
	Hong Kong		South Korea		U.S.	
Attribute	Without Block-Chain	With Block-Chain	Without Block-Chain	With Block-Chain	Without Block-Chain	With Block-Chain
SCTr *	4.28	5.44	7.93	8.79	11.74	15.26
Temp_pos **	7.40	7.52	11.75	11.80	27.05	30.26
Temp_neg ***	1.57	1.68	1.81	1.86	0.31	3.51
USc	14.71	14.68	10.71	12.76	51.85	50.44
SKc	14.95	14.93	20.99	23.04	4.23	2.82
AUSc	18.65	18.63	14.32	16.38	18.46	17.05
CANc	16.61	16.59	12.91	14.97	26.49	25.08
BRZc	10.35	10.33	6.24	8.29	5.64	4.23

* WTP for beef supply chain traceability vs. beef not supply chain traceable. ** WTP expressed relative to beef with temperature history unknown. *** WTP for beef from respective origin relative to origin unknown

## Data Availability

The data used to support the findings of this study can be made available by the corresponding author upon request.

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
