# Peer review of "The Intrinsic and Instrumental Values of Blockchain to Provide Beef Traceability in Hong Kong, South Korea, and the United States"

_foods, 2023, doi:10.3390/foods12234209_

Round 1
Reviewer 1 Report
Comments and Suggestions for Authors
The Publication "The Intrinsic and Instrumental Values of Blockchain to Provide Beef Traceability in Hong Kong, South Korea, and the United States" can be considered innovative due to the use of modern technology in the context of tracking food quality and the analysis of consumer preferences and values ​​in various global markets.
The authors of the publication conducted a well-executed survey and provided a clear and insightful description of their results and discussion.
I believe that everything the authors intended to prove by conducting the survey was achieved.
Comments on the Quality of English Language
Minor editing of English language required.
Author Response
Thank you for your comments. I have now revised the English in this paper.
Reviewer 2 Report
Comments and Suggestions for Authors
Comment to authors
- General comments: The authors used a questionnaire survey to determine the values of Blockchain to Provide Beef Traceability in Hong Kong, South Korea, and the United States. The English language usage is fine and the findings are good. However, there are a few methodological issues that need to be satisfactorily addressed before the manuscript can reach a publishable value.
- Lines 28-160: What is the difference between “introduction” and “background” that warranted splitting the supposed one section into two different sections?
Materials and methods:
- Can the authors provide a link to the online questionnaire used in this survey or attach the soft copy as a supplementary file to this manuscript?
- Was the questionnaire qualitatively and statistically validated and pre-tested before deployment, to ensure the reliability of the instrument of data collection and allay fears that the data collected are free of bias?
The authors may refer to these manuscripts for guidance
(1) COVID-19 Vaccine Hesitancy and Determinants of Acceptance among Healthcare Workers, Academics and Tertiary Students in Nigeria. Vaccines (Basel). 2022 Apr 15;10(4):626. doi: 10.3390/vaccines10040626.
(2) Mpox in Nigeria: Perceptions and knowledge of the disease among critical stakeholders-Global public health consequences. PLoS One. 2023 Mar 30;18(3):e0283571. doi: 10.1371/journal.pone.0283571.
- If the questionnaire were statistically validated, what was the Cronbach's Alpha value or reliability coefficient obtained?
- Did the authors obtain institutional ethical approval to conduct this survey? If so, kindly provide the link to the approval or the approval number as well as the granting body to this manuscript
- Did the authors obtain informed consent, to partake in the survey from the respondents, in line with the Helsinki Declaration of 2013?
- Figures 1 -3: Are these really figures or chart boxes?
- Table titles and Figure legends: These are consistently not descriptive enough. Tables and Figures should be able to stand alone such that the readership should be able to fully understand the information being conveyed without making reference to the body of the manuscript.
The English is fine in my view. Minor (inhouse) editing may be required
Author Response
General comments: The authors used a questionnaire survey to determine the values of Blockchain to Provide Beef Traceability in Hong Kong, South Korea, and the United States. The English language usage is fine and the findings are good. However, there are a few methodological issues that need to be satisfactorily addressed before the manuscript can reach a publishable value.
- Lines 28-160: What is the difference between “introduction” and “background” that warranted splitting the supposed one section into two different sections?
Response: Thank you for your question. In the “introduction” section, we emphasize the importance of this study. Then in the “background”, we discuss the literature review to demonstrate the contribution of this research, including topics such as blockchain applications in the food traceability system, the impact of blockchain applications on the food supply chain in terms of firm performance, consumer perceptions and the demand for blockchain applications, and consumer preferences and willingness to pay (WTP) for beef traceability. We decided to split the “introduction” and “background” sections to convey a clearer message to readers of Foods. However, if you believe combining the two sections would be better and clearer for the readers, we are open to merging them into one section. You can now find the combined “introduction” section.
- Materials and methods: Can the authors provide a link to the online questionnaire used in this survey or attach the soft copy as a supplementary file to this manuscript?
Response: The survey link has expired since the survey was completed. I have now attached the soft copy as a supplementary file.
- Was the questionnaire qualitatively and statistically validated and pre-tested before deployment, to ensure the reliability of the instrument of data collection and allay fears that the data collected are free of bias?
The authors may refer to these manuscripts for guidance
(1) COVID-19 Vaccine Hesitancy and Determinants of Acceptance among Healthcare Workers, Academics and Tertiary Students in Nigeria. Vaccines (Basel). 2022 Apr 15;10(4):626. doi: 10.3390/vaccines10040626.
(2) Mpox in Nigeria: Perceptions and knowledge of the disease among critical stakeholders-Global public health consequences. PLoS One. 2023 Mar 30;18(3):e0283571. doi: 10.1371/journal.pone.0283571.
Response: We wrote the original survey in WORD and in English and pretested with colleagues and students. Then, the survey was translated into the respective languages, programmed in Qualtrics, and again checked for understanding and readability. In terms of statistical validity, we obtained over 500 responses in each country. This would yield a sampling error of +/- 4.4% on a single binary choice question. However, note that for the choice experiment, each respondent answered 10 choice questions, which would yield a sampling error of about 1.4%. The Choice Experiment was developed using well-established methodologies, and the approach has shown high level of external validity (e.g., Lusk and Brooks, 2010; Swait and Andrews). As described in the text, the Choice Experiment questions were designed based on a Bayesian optimal (or D-optimal) experimental design, which was constructed to minimize the standard errors of a multinomial logit choice model. Some of this is described on page 7 of the revised text.
- If the questionnaire were statistically validated, what was the Cronbach's Alpha value or reliability coefficient obtained?
Response: Cronbach's Alpha is normally calculated when constructing multi-item scales, where a single construct is measured using multiple individual likert-type questions. There is no equivalent for Choice Experiments, where the aim is to estimate an underlying latent random utility function. The fact that the attribute values are statistically signification provides a measure of internal validity insofar as showing respondents reacted to changes in the attributes (or stimuli) in a non-random fashion, and in a manner consistent with our expectations or hypotheses.
- Did the authors obtain institutional ethical approval to conduct this survey? If so, kindly provide the link to the approval or the approval number as well as the granting body to this manuscript
Response: Ethical review and approval were waived for this study because recorded information cannot readily identify the subjects, and any disclosure of responses outside of the research would not reasonably place subjects at the risk. However, we obtain ethical consent from the participants before they take the survey, ensuring that they are fully informed and aware of the research process. We’ve included ‘Institutional Review Board Statement’ on page 20.
- Did the authors obtain informed consent, to partake in the survey from the respondents, in line with the Helsinki Declaration of 2013?
Response: Yes, we obtain informed consent from the respondents. Additionally, we provide about 3 dollars in incentives to those who complete the survey. We’ve included ‘Informed Consent Statement’ on page 20.
- Figures 1 -3: Are these really figures or chart boxes?
Response: Yes, you are right. These figures align with the information provided by the respondents in the survey.
- Table titles and Figure legends: These are consistently not descriptive enough. Tables and Figures should be able to stand alone such that the readership should be able to fully understand the information being conveyed without making reference to the body of the manuscript.
Response: Thank you for your comments. Now we’ve revised the table titles and figure legends to stand alone more effectively. You can find them on pages 6, 8, 9, 17, and 18.
Reviewer 3 Report
Comments and Suggestions for Authors
This article seems very useful in the literature because it deals with the issue of food traceability. Unfortunately, to date, not for all foods there is a system that allows you to know all the stages, from production to the consumer. For some foods, such as fish, meat and fresh food in general, it is important to have this information in particular thanks to blockchain technology that makes the entire supply chain transparent, otherwise you risk eating a deteriorated product.
In general, the work is well structured: Introduction, Materials and Methods, Results, Discussions and Conclusions. Before being accepted for publication it is important to perform major revisions:
- In the Background section the works that used the blockchain have been reported but there is no specific reference for the various foods. Since the article talks about meat, it would be important to include some references and work that talk about the traceability of meat, using the blockchain. In addition, some examples linked to other food chains traced with blockchain are requested;
- In the Materials and Methods section, there is talk of 1579 consumers responding to the questionnaire. Is this number representative of the Korean, US and Hong Kong population? Explain why the number of consumers per country is so low compared to the number of population;
- the results are clear;
- In the discussion it is important to add a few references to support your results. In particular, explain why blockchain is so important compared to other tracking systems. Compare other jobs done with yours;
- In the conclusion, it is important to understand whether this is just a one-off job or whether you foresee future work that will deepen the blockchain in the food sector.
- Recheck the bibliographic formatting.
Congratulations for the good work!
Author Response
This article seems very useful in the literature because it deals with the issue of food traceability. Unfortunately, to date, not for all foods there is a system that allows you to know all the stages, from production to the consumer. For some foods, such as fish, meat and fresh food in general, it is important to have this information in particular thanks to blockchain technology that makes the entire supply chain transparent, otherwise you risk eating a deteriorated product.
In general, the work is well structured: Introduction, Materials and Methods, Results, Discussions and Conclusions. Before being accepted for publication it is important to perform major revisions:
- In the Background section the works that used the blockchain have been reported but there is no specific reference for the various foods. Since the article talks about meat, it would be important to include some references and work that talk about the traceability of meat, using the blockchain. In addition, some examples linked to other food chains traced with blockchain are requested;
Response: In the ‘Introduction’ section, we’ve included additional literature reviews concerning blockchain application cases in meat and other food traceability system. You can find it on page 2 and 3.
- In the Materials and Methods section, there is talk of 1579 consumers responding to the questionnaire. Is this number representative of the Korean, US and Hong Kong population? Explain why the number of consumers per country is so low compared to the number of population;
Response: As stated in the manuscript, comparing demographic characteristics to census-level data is crucial to assess representativeness. However, our primary concern lies in the sample’s representation of beef steak consumers. Unfortunately, census-level information specific to beef consumers is unavailable. To address this, we ensured representativeness by excluding respondents who indicated they do not consume beef.
In terms of statistical validity, we obtained over 500 responses in each country. This would yield a sampling error of +/- 4.4% on a single binary choice question. However, note that for the choice experiment, each respondent answered 10 choice questions, which would yield a sampling error of about 1.4%. The Choice Experiment was developed using well-established methodologies, and the approach has shown high level of external validity (e.g., Lusk and Brooks, 2010; Swait and Andrews). As described in the text, the Choice Experiment questions were designed based on a Bayesian optimal (or D-optimal) experimental design, which was constructed to minimize the standard errors of a multinomial logit choice model. Some of this is described on page 7 of the revised text.
- the results are clear;
Response: Thank you.
- In the discussion it is important to add a few references to support your results. In particular, explain why blockchain is so important compared to other tracking systems. Compare other jobs done with yours;
Response: Thank you for the comments. We initially highlighted the necessity of blockchain technology in the food traceability system in the ‘introduction’ section through literature review. To provide a clearer message, we have now incorporated this discussion into the ‘discussion’ section.
- In the conclusion, it is important to understand whether this is just a one-off job or whether you foresee future work that will deepen the blockchain in the food sector.
Response: As mentioned in the text, blockchain applications in the food industry are rapidly expanding. According to Market and Market, the estimated size of the blockchain market for agriculture and the food supply chain was $133.4 million in 2020, and it is projected to reach $948.6 million in 2025. This study explores consumers’ preferences and willingness to pay (WTP) for beef traceability information, including details related to the supply chain, temperature history, and country of origin. We then determined how WTP for the three types of information varied when provided through blockchain technology.
Based on these results, we propose practical implementations for applying blockchain technology to beef traceability systems. Additionally, in the ‘Conclusion’ section, we discuss avenues for further research on blockchain applications in the food industry.
- Recheck the bibliographic formatting.
Response: Now we’ve revised the bibliographic formatting.
Round 2
Reviewer 3 Report
Comments and Suggestions for Authors
The article was improved following revisions that were suggested. In my opinion it can be published, as it is a topic of increasing importance in recent years and expanding. Congratulations.
Author Response
Thank you for your comments.